# Developmental stage-specific spontaneous activity contributes to callosal axon projections

**Yuta Tezuka[1†], Kenta M Hagihara[2†‡], Kenichi Ohki[2,3,4,5,6], Tomoo Hirano[1], Yoshiaki Tagawa[1,6,7*]**

[1]Department of Biophysics, Kyoto University Graduate School of Science, Kyoto, Japan; [2]Department of Molecular Physiology, Kyushu University Graduate School of Medical Sciences, Fukuoka, Japan; [3]Department of Physiology, The University of Tokyo School of Medicine, Tokyo, Japan; [4]International Research Center for Neurointelligence (WPI-IRCN), The University of Tokyo School of Medicine, Tokyo, Japan; [5]Institute for AI and Beyond, The University of Tokyo School of Medicine, Tokyo, Japan; [6]CREST, Japan Science and Technology Agency, Saitama, Japan; [7]Department of Physiology, Graduate School of Medical and Dental Sciences, Kagoshima University, Kagoshima, Japan

**\*For correspondence:**
tagawa@m.kufm.kagoshima-u.ac.jp

[†]These authors contributed equally to this work

**Present address:** [‡]Allen Institute for Neural Dynamics, Seattle, United States

**Competing interest:** The authors declare that no competing interests exist.

**Abstract** The developing neocortex exhibits spontaneous network activity with various synchrony levels, which has been implicated in the formation of cortical circuits. We previously reported that the development of callosal axon projections, one of the major long-range axonal projections in the brain, is activity dependent. However, what sort of activity and when activity is indispensable are not known. Here, using a genetic method to manipulate network activity in a stage-specific manner, we demonstrated that network activity contributes to callosal axon projections in the mouse visual cortex during a 'critical period': restoring neuronal activity during that period resumed the projections, whereas restoration after the period failed. Furthermore, in vivo Ca$^{2+}$ imaging revealed that the projections could be established even without fully restoring highly synchronous activity. Overall, our findings suggest that spontaneous network activity is selectively required during a critical developmental time window for the formation of long-range axonal projections in the cortex.

## Editor's evaluation

This study extends previous findings on the activity-dependent development of the long-range axonal connections between the cortical hemispheres. In particular, it shows that neuronal activity strongly influences the development of these axons during a particular developmental period and it provides convincing evidence that specific activity patterns are required for the proper development of these long-range connections.

## Introduction

Neuronal activity plays a role in the formation of neural circuits in the brain (*Katz and Shatz, 1996*; *Spitzer, 2006*; *Ackman and Crair, 2014*). The roles of sensory-driven and spontaneously generated neuronal activity in circuit formation are well documented in the mammalian visual system (*Wong, 1999*; *Huberman et al., 2008*). Retinal ganglion cells (RGCs), output neurons in the retinal circuit, exhibit spontaneously generated, highly correlated neuronal activity (called retinal waves) before photoreceptor cells develop (*Galli and Maffei, 1988*; *Meister et al., 1991*; *Wong et al., 1993*).

Such activity is transmitted to the lateral geniculate nucleus and the superior colliculus, where it plays an instructive role in the formation and reorganization of neuronal connections (*Penn et al., 1998*; *Stellwagen and Shatz, 2002*; *Pfeiffenberger et al., 2005*; *Chandrasekaran et al., 2005*; *Hooks and Chen, 2006*). The connectivity once formed is further sculpted by sensory-driven neuronal activity after visual inputs are available (*Hooks and Chen, 2006*).

The developing cerebral cortex also exhibits robust spontaneous network activity, ranging from highly synchronous to less correlated patterns (*Siegel et al., 2012*). This is partly generated intra-cortically and partly originates from retinal waves (*Siegel et al., 2012*; *Gribizis et al., 2019*; *Smith et al., 2018*). In rodents, a highly synchronous pattern of spontaneous neuronal activity emerges during early postnatal periods when cortical neurons still undergo maturation and circuit formation (*Garaschuk et al., 2000*; *Allène et al., 2008*; *Yang et al., 2009*; *Siegel et al., 2012*). Such activity indicates a heterogeneous traveling pattern in the developing cortex (*Garaschuk et al., 2000*; *Ackman et al., 2012*; *Hagihara et al., 2015*). Subsequently, a more desynchronized pattern becomes dominant (*Rochefort et al., 2009*; *Golshani et al., 2009*). Based on findings in the visual system and other systems, these patterns of activity have been thought to play important roles in the development of cortical circuits (*Khazipov and Luhmann, 2006*; *Blankenship and Feller, 2010*; *Winnubst et al., 2015*; *Hagihara et al., 2015*; *Nakazawa et al., 2020*). However, their role in many aspects of cortical circuit formation remains to be fully elucidated.

The callosal axon projection, a long-range axonal projection connecting the two cortical hemispheres, is essential for integrating information processed in the hemispheres (*Hubel and Wiesel, 1967*). It is important for higher cognitive functions and its alterations have been noted in patients with developmental disorders (*Paul, 2011*). It has been a good model for studying activity-dependent mechanisms of cortical circuit formation (*De León Reyes et al., 2020*). Callosal axons derived from layer 2/3 callosal projection neurons in one hemisphere project to the other hemisphere through several successive stages by postnatal day 15 (P15) in mice (*Mizuno et al., 2007*; *Wang et al., 2007*; *Tagawa and Hirano, 2012*). This axonal projection is activity-dependent: several groups, including ours, have previously shown that exogenous expression of Kir2.1 (an inward rectifying potassium channel), a genetic method to reduce neuronal activity (*Johns et al., 1999*), impairs callosal axon projection (*Mizuno et al., 2007*; *Wang et al., 2007*; *Suárez et al., 2014*; *Rodríguez-Tornos et al., 2016*). Interestingly, when we expressed Kir2.1 in only a small number of neurons and examined the cell-autonomous effect of activity reduction on their axonal projections, we observed that the effect was less than that when we expressed Kir2.1 in a larger number of neurons (*Mizuno et al., 2010*). We also found that Kir2.1 expression in a large number of neurons reduced not only the activity of individual neurons (*Mizuno et al., 2007*) but also spontaneous network activity in the cortex during early postnatal periods (*Hagihara et al., 2015*). As mentioned above, the pattern of spontaneous cortical network activity changes during the period when callosal axons undergo maturation. These results led us to assume that there might be a specific temporal window in which cortical network activity is indispensable for the formation of long-range axonal projections of callosal neurons.

In this report, we sought to determine what sort of activity and when activity is indispensable for callosal axon projections. To this end, we used genetic methods to reduce and restore spontaneous network activity in a stage-specific manner in the developing mouse cortex.

## Results

### Callosal axons require neuronal activity during P6-15 for their projection

In the visual cortex, callosal axons project densely to a narrowly restricted region at the border between the primary and secondary visual cortex, in which they terminate primarily in layer 2/3 and less so in layer 5 (*Olavarria and Montero, 1984*; *Mizuno et al., 2007*; *Figure 1—figure supplement 1*). This region- and lamina-specific projection pattern forms by P15 in the mouse (*Mizuno et al., 2007*). Our group, as well as others, previously showed that Kir2.1 expression disturbs callosal axon development primarily in the second postnatal week (*Tagawa and Hirano, 2012*). We also reported that Kir2.1 expression strongly reduced spontaneous cortical network activity at P9-10 and P13-14 (*Hagihara et al., 2015*). These results suggest that network activity around the second postnatal week is involved in the formation of callosal axon projections. In the current study, we sought to confirm this

by conducting 'rescue' experiments, in which we attempted to restore the activity during P6-15 and asked whether such restored activity could recover the axonal projections.

To control spontaneous network activity in a stage-specific manner during early postnatal periods, we used the Tet-off system to express Kir2.1. In this system, Kir2.1 was expressed without doxycycline (Dox) treatment, and Kir2.1 expression was suppressed in response to Dox treatment (*Figure 1A*). Using in utero electroporation, we transfected layer 2/3 cortical neurons with two expression vectors (pTRE-Tight2-Kir2.1 and pCAG-tTA2ˢ) for the expression of Kir2.1, together with an RFP expression vector (pCAG-TurboRFP) for labeling callosal axon projections (*Figure 1*; *Mizuno et al., 2007*; *Hagihara et al., 2015*).

To test the feasibility of the Tet-off system, we divided the electroporated mice into two groups. We reared the first group of mice without Dox treatment throughout development (*n*=7 mice). Under this condition, we observed that RFP-labeled callosal axon projections were impaired at P15 (*Figure 1B*, *Figure 1—figure supplement 1*), a pattern similar to that observed when we expressed both Kir2.1 and a fluorescent protein under the control of the CAG promoter (*Mizuno et al., 2007*; *Figure 1—figure supplement 1*). This suggests that Kir2.1 expression using the Tet-off system is as effective as that using the CAG promoter. We then reared the second group with Dox throughout development (from E15 to P15: see Methods: *n*=8 mice). Under this condition, we observed that RFP-labeled callosal axons projected normally to the contralateral cortex in a region- and lamina-specific manner at P15 (*Figure 1B*, *Figure 1—figure supplement 1*). This projection pattern was similar to the pattern observed when we expressed only a fluorescent protein (i.e. normal pattern of callosal projections; *Mizuno et al., 2007*; *Figure 1—figure supplement 1*), suggesting that Dox treatment effectively suppressed Kir2.1 expression throughout development.

The aim of this study was to determine the role of neuronal activity during P6-15. Therefore, we used the same set of plasmids for in utero electroporation and began Dox treatment during the postnatal period. Dox treatment for 4 days was almost enough to suppress gene expression in the Tet-off system (*Figure 1—figure supplement 2*; *Figure 1—figure supplement 3*; regarding the apparent low-level expression of Kir2.1EGFP after Dox treatment, please see the legend of *Figure 1—figure supplement 3*). We started Dox treatment from P6 and continued it until P15 (*n*=10 mice). Under this condition, we observed that RFP-labeled callosal axons reached the innervation area in the contralateral cortex and ramified in a laminar specific manner at P15 (*Figure 1B, C*). This projection pattern appeared comparable to the normal pattern of callosal axon projections (*Figure 1B, C*). Because reaching and ramifying in layer 2/3 is a critical developmental step for the formation of callosal axon projections (*Mizuno et al., 2007*; *Wang et al., 2007*; *Tagawa and Hirano, 2012*), we quantified the strength of RFP signals in layer 2/3. Quantitative analyses suggested that the density of RFP-labeled callosal axons arriving and ramifying in the target cortical layer (see Methods) was comparable between the Dox P6-15 and Dox E15-P15 groups (p=0.96, Tukey–Kramer test; *Figure 1D*), and that it was greater in the Dox P6-15 group than in the no Dox group (p=$9.0 \times 10^{-4}$, one-way ANOVA; p=$2.3 \times 10^{-3}$, Tukey–Kramer test; *Figure 1D*). Analyses of the width of callosal axon innervation zone, as well as densitometric line scans across all cortical layers, suggest that some aspects of callosal projections might be partially recovered in Dox P6-15 mice (*Figure 1—figure supplement 4*). These results suggest that Dox treatment at P6-15, which likely shut off Kir2.1 expression from P8-10 to P15, was effective for the formation of callosal axon projections, implying that callosal axons require neuronal activity during P6-15 to form their projections.

An important role of neuronal activity in P10-15 was suggested by another experiment using the DREADD (designer receptors exclusively activated by designer drugs) method. DREADD is a method to increase or decrease neuronal activity using a nonendogenous ligand clozapine-N-oxide (CNO; *Alexander et al., 2009*). Activation of hM3DGq, one of the artificial G-protein-coupled receptors used in DREADD technology, in hippocampal neurons causes depolarization of the membrane potential, resulting in an increase in neuronal activity (*Alexander et al., 2009*). We transfected layer 2/3 cortical neurons with the expression vector pCAG-hM3DGq, together with the pCAG-TurboRFP and pCAG-Kir2.1 plasmids, by in utero electroporation at E15 (*Figure 2A*). We observed that the electroporated mice with daily CNO injections from P10 to P14 (*n*=11 mice) exhibited a region- and lamina-specific projection pattern of RFP-labeled callosal axons in the contralateral cortex (*Figure 2B*). Neither of the two control groups of mice (*n*=10 mice for Kir2.1+hM3 DGq with saline injection P10-14, *n*=4 mice for Kir2.1 with CNO injection P10-14) exhibited such axonal projections (*Figure 2B*,

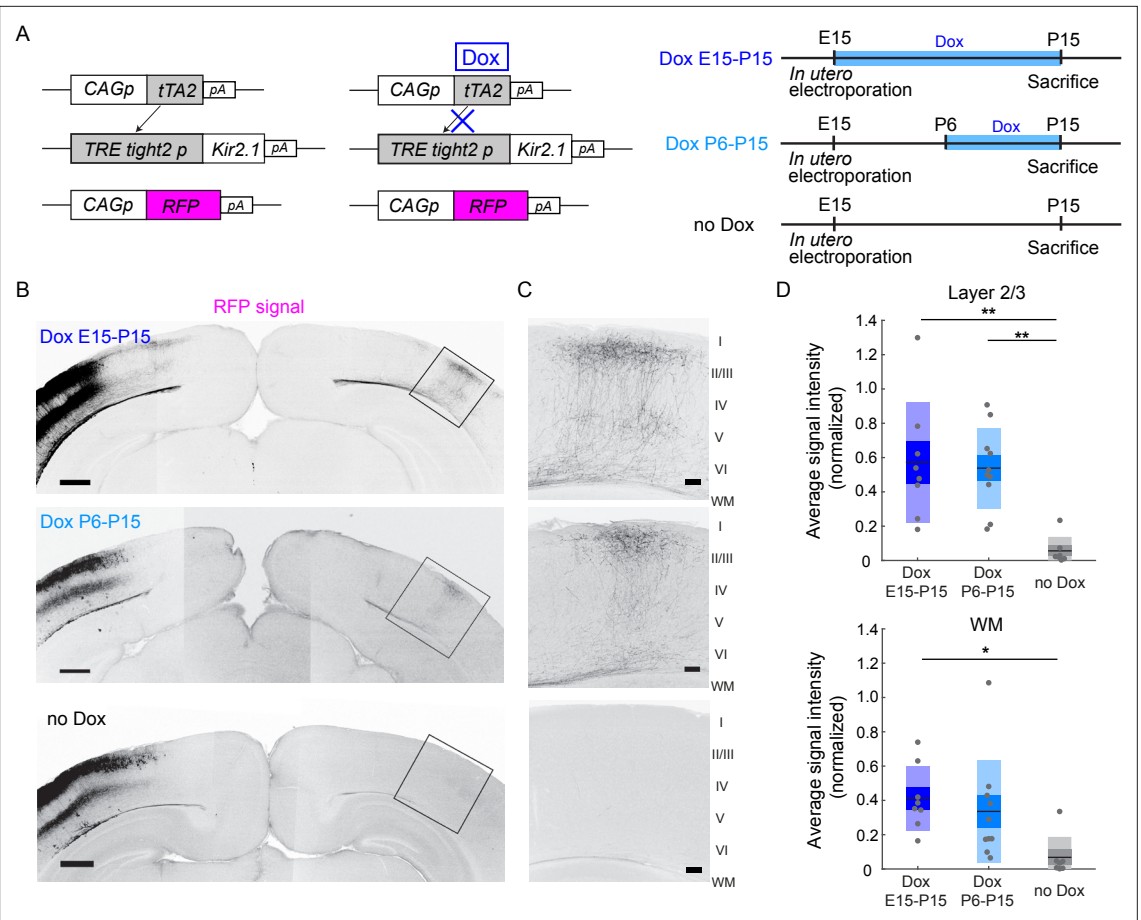

**Figure 1.** Restoration of neuronal activity in the second postnatal week recovers callosal axon projections. (**A**) Left, plasmids designed for temporally controlled expression of Kir2.1. Right, experimental timeline. Kir2.1 is a genetic tool to reduce neuronal activity. Transactivator tTA2ˢ is expressed under the control of the CAG promoter (CAGp). Without doxycycline (Dox), tTA2ˢ binds to the TRETight2 promoter (TRETight2p), and Kir2.1 expression is induced (left). With Dox administration, tTA2ˢ cannot bind to the TRETight2 promoter, and Kir2.1 expression is suppressed (right). pA: polyA signal. (**B**) Coronal sections through the P15 cerebral cortex show the distribution of neurons expressing fluorescent proteins on the electroporated side and their axonal projections on the other side. Top: 'Dox E15-P15' condition where Dox was administered from E15 to P15; thus, Kir2.1 expression was suppressed throughout development. Middle: 'Dox P6-15' condition where Dox was administered from P6 to P15; thus, Kir2.1 expression was suppressed in the second postnatal week. Bottom: 'no Dox' condition where Kir2.1 is expressed throughout development. Scale bars, 500 μm. (**C**) Boxed regions in (**B**). WM: white matter. Scale bars, 100 μm. (**D**) Top, average signal intensity of fluorescently labeled callosal axons in layers 1–3. 'Dox E15-P15' group: n=8 sections from eight mice. 'Dox P6-15' group: n=10 sections from 10 mice. 'no Dox' group: n=7 sections from seven mice. \*\*p<0.01 by Tukey–Kramer test. Bottom, average signal intensity of fluorescently labeled callosal axons in the white matter. \*p<0.05 by Tukey–Kramer test.

The online version of this article includes the following source data and figure supplement(s) for figure 1:

**Source data 1.** Source data for *Figure 1D*.

**Figure supplement 1.** Callosal axon projections with or without activity manipulation.

**Figure supplement 2.** Validation of the Tet-off system with fluorescent proteins.

**Figure supplement 3.** Validation of the Tet-off system with Kir2.1EGFP.

**Figure supplement 3—source data 1.** Source data for *Figure 1—figure supplement 3D and E*.

**Figure supplement 4.** The width and laminar distribution of callosal axon projections.

**Figure supplement 4—source data 1.** Source data for *Figure 1—figure supplement 4*.

data not shown). Quantitative analyses suggested a partial but significant 'rescue' effect of hM3DGq expression with CNO injections against the effect of Kir2.1 expression on callosal axon projections (*Figure 2C*). These results suggest that the resumption of neuronal activity during P10-15 is effective for the formation of callosal axon projections.

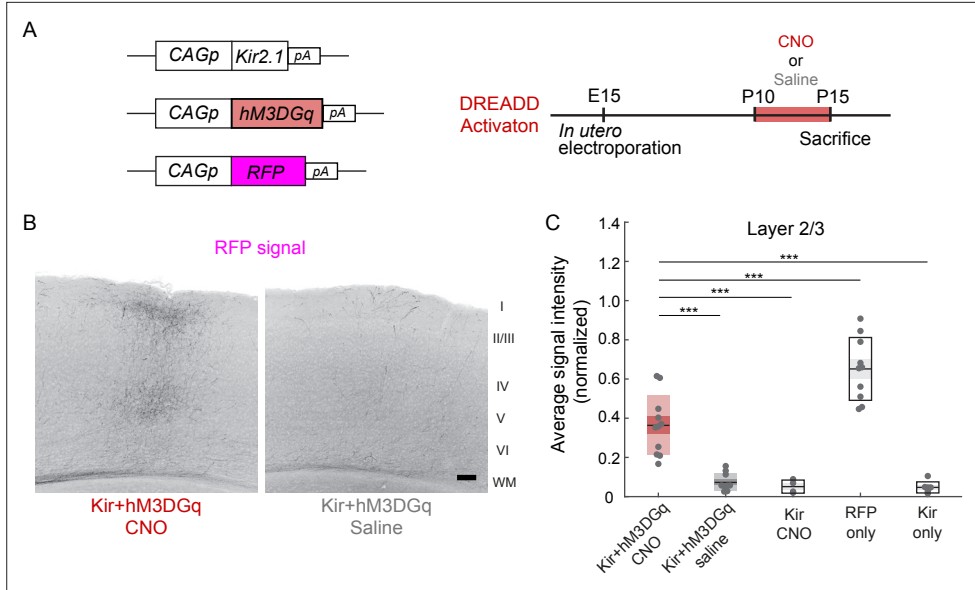

**Figure 2.** Effect of activity restoration on callosal axon projections using designer receptors exclusively activated by designer drugs (DREADD). (**A**) Experimental design and timeline. Kir2.1 expression and hM3DGq expression plasmids were transfected at E15, clozapine-N-oxide (CNO) or saline was injected daily from P10 to P14, and brains were fixed at P15. (**B**) Fluorescently labeled callosal axon projections in the P15 cortex. Scale bar, 100 µm. (**C**) Average signal intensity of fluorescently labeled callosal axons in layers 1–3. 'Kir + hM3 DGq with CNO injections' group; *n*=11 sections from 11 mice. 'Kir + hM3 DGq with saline injections' group; *n*=10 sections from 10 mice. 'Kir with CNO injections' group: *n*=4 sections from four mice. 'RFP' group (CAG promoter [pCAG]-TurboRFP was electroporated, no injection was performed): *n*=10 sections from 10 mice. 'Kir' group (pCAG-Kir2.1 was electroporated, no injection was performed): *n*=7 sections from seven mice. ***p<0.001 by Tukey–Kramer test.

The online version of this article includes the following source data for figure 2:

**Source data 1.** Source data for *Figure 2*.

## A critical period for the formation of callosal projections

Callosal axons from Kir2.1-expressing neurons arrive in the white matter (WM) of the target innervation areas around P5, the same time as normal callosal axons (*Mizuno et al., 2007*). Afterward, they exhibit retarded and eventually stalled growth and development. As shown above, callosal axons can form region- and lamina-specific projection patterns when their activity is resumed from P6. How long do they retain the ability to grow into the target cortical areas and make lamina-specific branches after reaching the target innervation areas around P5? We found that the region- and lamina-specific projection pattern of callosal axons could not be recovered when Dox treatment was started later than P9. We performed in utero electroporation with the same set of plasmids as before (*Figure 1A* and *Figure 3A*), started Dox treatment from P9 or P12, and then assessed callosal axon projections at P18 or P21 (*n*=9 mice for Dox treatment P918: *n*=10 mice for Dox treatment P12-21). Contrary to the result of Dox treatment during P6-15, callosal axons were not present in the target innervation areas in either group of mice (*Figure 3B*). The extent of callosal axons arriving and ramifying in the target cortical layer (see Methods) was significantly lower in the Dox P9-18 and Dox P12-21 groups than in the Dox P6-15 group (p=$2.1 \times 10^{-5}$, one-way ANOVA; Dox P9-18 vs Dox P6-15, p=$1.0 \times 10^{-4}$; Dox P12-21 vs Dox P6-15, p=$9.0 \times 10^{-5}$, Tukey–Kramer test; *Figure 3C*). These observations suggest that callosal axons retain the ability to grow into the target innervation areas and make lamina-specific branches only for a limited period during development.

## Network activity contributing to callosal axon projections

It has been shown that spontaneous network activity frequently occurs in the early postnatal cerebral cortex. Such network activity can be classified as a highly synchronous pattern (H events) or a less correlated pattern (L events) (*Siegel et al., 2012*). In the visual cortex, these events are mechanistically distinct (*Siegel et al., 2012*): H events are mostly of cortical origin, whereas L events are driven

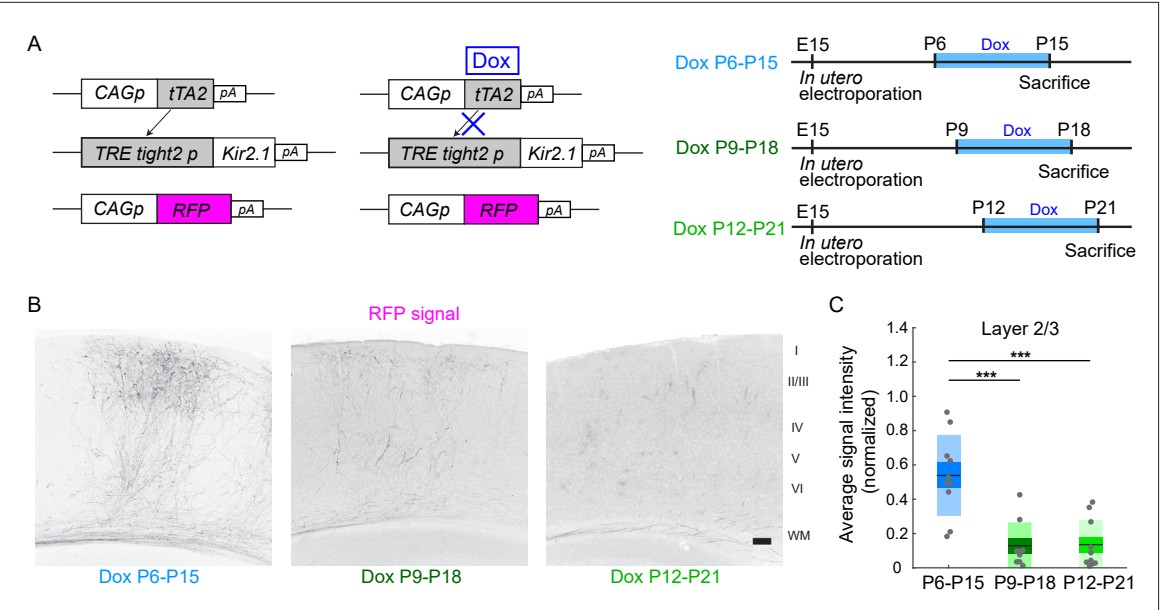

**Figure 3.** A critical period for restoration of callosal axon projections. (**A**) Experimental design and timeline. CAG promoter (pCAG)-tTA2s and TRETight2 promoter (pTRETight2)-Kir2.1 plasmids were transfected at E15, and doxycycline (Dox) was administered from P6 to P15, from P9 to P18, or from P12 to P21. (**B**) Fluorescently labeled callosal axon projections in the P15 cortex. Scale bar, 100 µm. (**C**) Average signal intensity of fluorescently labeled callosal axons in layers 1–3. 'Dox P6-15' group: $n$=10 sections from 10 mice. 'Dox P9-P18' group: $n$=9 sections from nine mice. 'Dox P12-P21' group: $n$=10 sections from 10 mice. ***$p$<0.001 by Tukey–Kramer test.

The online version of this article includes the following source data for figure 3:

**Source data 1.** Source data for *Figure 3C*.

by activity in the retina. They may have distinct roles in cortical circuit formation (*Cheyne et al., 2019*; *Wosniack et al., 2021*). To gain insight into what pattern(s) of network activity contribute to activity-dependent axonal projections, we recorded spontaneous neuronal activity in V1 at P13 using in vivo two-photon $Ca^{2+}$ imaging with a calcium indicator OGB1 (*Figure 4A*). Consistent with a previous study (*Hagihara et al., 2015*), control mice (in which the fluorescent protein FP635 was electroporated: $n$=5 mice) exhibited frequent spontaneous network events during 10-minute recording sessions (*Figure 4B, C*), whereas Kir2.1-expressing mice exhibited substantially fewer network events ($n$=5 mice, *Figure 4E*). Notably, both Kir2.1-positive and Kir2.1-negative neurons were suppressed. We classified these network events based on a similar criterion to that used in a previous study (*Siegel et al., 2012*, see Methods): H events, highly synchronous network activity with a participation rate >60%; L events, less correlated events with a participation rate of 60–20%. We observed both patterns of network activity in control mice at P13 (*Figure 4F*) (H events, 1.70±0.36 events/min; L events, 2.16±0.27 events/min), and Kir2.1 expression significantly reduced both of them (*Figure 4F*) (H events, 0.21±0.10 events/min; L events, 0.62±0.09 events/min; H events, $p$=1.3 × 10$^{-3}$; L events, $p$=4.9 × 10$^{-4}$ by one-way ANOVA; H events, $p$=1.2 × 10$^{-3}$; L events, $p$=8.1 × 10$^{-4}$ by Tukey–Kramer test).

We then asked whether the network activity would be resumed in the condition where callosal axon projections were recovered (i.e. Kir2.1 expression was turned off beginning from P6). We started Dox treatment on P6 and recorded spontaneous neuronal activity in the electroporated mice at P13 ($n$=5 mice). We found that Dox treatment from P6 resumed cortical network activity at P13 (*Figure 4D*). Unexpectedly, this treatment almost recovered L events, whereas H events were not recovered (H events, 0.66±0.21 events/min; L events, 2.01±0.30 events/min; *Figure 4G*). L events may make a larger contribution to the formation of callosal axon projections, although the involvement of H events or the total frequency of activity cannot be ruled out. Collectively, our observations suggest that sufficiently high activity during P6-15 plays a role in the formation of region- and lamina-specific projection patterns of callosal axons.

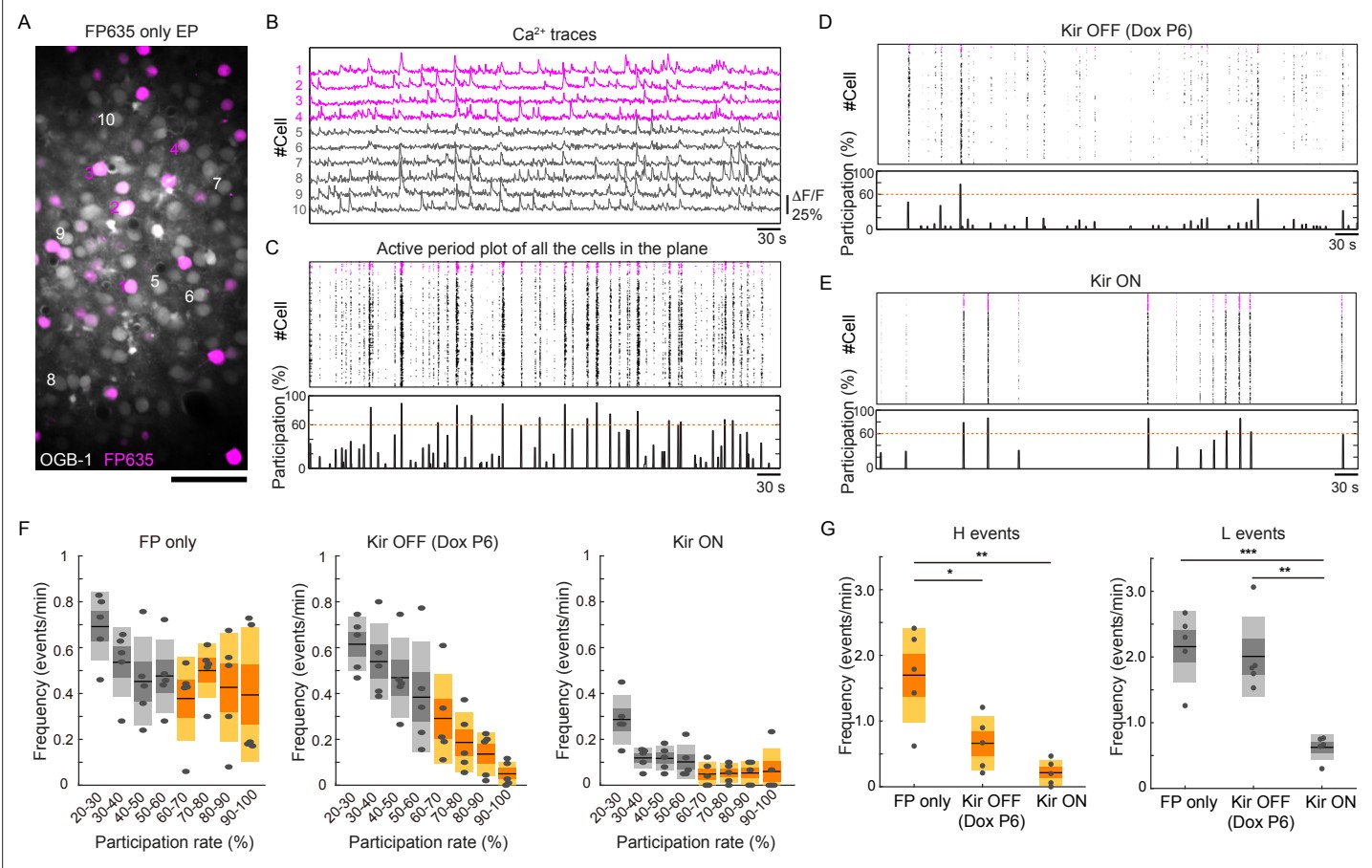

**Figure 4.** Partial restoration of synchronous activity revealed by in vivo two-photon $Ca^{2+}$ imaging. (**A**) A representative image of OBG1-loaded (gray) and FP635-expressing (magenta) neurons in the P13 mouse visual cortex obtained by two-photon microscopy. Scale bar: 50 μm. Numbers shown in the image correspond to the cell numbers shown in panel (**B**). (**B**) $Ca^{2+}$ traces from representative FP635+ (#1–4) and FP635− (#5–10) neurons. Obtained from the same recording session as shown in the bottom panel. (**C**) Top: an active period plot of all identified FP635+ (magenta) and FP635− (black) neurons in the top image. Bottom: corresponding participation rate. Events with a participation rate above 60% were regarded as H events. (**D**) A representative active period plot and corresponding participation rate from a Tet-Kir2.1 with doxycycline (Dox) mouse. Because of Dox treatment from P6, Kir2.1 expression was suppressed from P6: Kir OFF. (**E**) A representative active period plot and corresponding participation rate from a Tet-Kir2.1 without Dox mouse. Because of the absence of Dox, Kir2.1 was expressed throughout development: Kir ON. (**F**) Histograms of synchronous activity frequency with different participation rates from FP635-electroporated (Left: FP only), Tet-Kir2.1 with Dox (Middle: Kir OFF Dox P6), and Tet-Kir2.1 without Dox (Right: Kir ON) mice. (**G**) Quantifications of H (>60%) and L (20–60%) events. FP only, five mice. Kir OFF (Dox P6), five mice. Kir ON, five mice. *p<0.05, **p<0.01, ***p<0.001 by Tukey–Kramer test.

The online version of this article includes the following source data for figure 4:

**Source data 1.** Source data for *Figure 4F and G*.

## Discussion

In this study, using a genetic method to reduce and restore spontaneous network activity in the early postnatal cerebral cortex, we demonstrated that spontaneous activity during P6-15 contributes to the formation of region- and lamina-specific projection patterns of callosal axons. Several previous studies have revealed important roles of spontaneous cortical activity in the fine-tuning of local circuits and neural function (*Winnubst et al., 2015*: *Hagihara et al., 2015*; *Wosniack et al., 2021*). Our findings add new evidence of the critical role of spontaneous network activity in cortical circuit formation and demonstrate that not only local circuits but also long-range axonal projections require spontaneous activity for their normal development.

A recent study showed that a developmental change in the firing mode of callosal projection neurons, regulated by transcription factor Cux1-driven Kv1 channel expression, was critically involved in activity-dependent callosal axon projections (*Rodríguez-Tornos et al., 2016*). Because they

assessed the firing property of neurons using in vitro slice preparations, what aspect(s) of activity in vivo contribute to axonal projection is not known. Our study extends their findings by demonstrating that spontaneous activity in vivo is critical for callosal axon projections.

The 'rescue' effect was only observed during P6-15 but not afterward (*Figure 3*). A similar developmental time window for the formation of axonal projections was reported in the olfactory system (*Ma et al., 2014*). In the previous study, we showed that callosal axons could reach the innervation area almost normally under activity reduction, and that the effects of activity reduction became apparent afterward (*Mizuno et al., 2007*). Callosal axons elaborate their branches extensively in P10-P15 (*Mizuno et al., 2010*), and axon branching is regulated by neuronal activity (*Matsumoto et al., 2016*). It is likely that activity is required for the processes of formation, rather than the maintenance of the connections already formed by P10, but the current study employed massive labeling of callosal axons which is not suited to clarify this. In addition, the restoration of activity in the Tet-off (*Figure 1*) or DREADD (*Figure 2*) experiment may not completely rescue the ramification pattern of individual axons. Single-axon tracing experiments (*Mizuno et al., 2010*; *Dhande et al., 2011*) would be required to clarify this. Nonetheless, our findings suggest that callosal axons retain the ability, or are permitted, to grow and make region- and lamina-specific projections in the cortex during a limited period of postnatal cortical development under an activity-dependent mechanism.

Patterned spontaneous activity in the developing neocortex can be classified as a highly synchronous pattern (H events) and a less correlated pattern (L events) (*Siegel et al., 2012*). What could be the roles of these patterns of activity in axonal projections? In retinothalamic and retinocollicular projections, network activity in the retina (retinal waves) is proposed to provide spatial information for axonal projections. It is proposed that two adjacent RGCs, which likely fire together in a spontaneous network event would project their axons to an adjacent location in the target tissue. On the other hand, two RGCs located distantly, or even in the left and right eyes, are unlikely to fire together, projecting axons to distinct locations (*Wong, 1999*). The highly synchronous network activity in the cortex recruits more than 60% of neurons in an imaging field, clearly exceeding the proportion of callosal projection neurons. Callosal projection neurons are intermingled with other types of cortical neurons that project axons intracortically (*Mitchell and Macklis, 2005*), suggesting that both the former and the latter contribute to H events once they occur. If the highly synchronous activity in the cortex exerts a similar role as retinal waves, it may provide topographic information to callosal axons and other axons projecting to other cortical regions at the same time.

We have shown that the projections could be established even without fully restoring highly synchronous activity (*Figure 4*). L events, but not H events, were present in P13 cortex after Dox treatment at P6. L events may be sufficient for the formation of callosal projections. Alternatively, any form of activity with certain level(s) (i.e. 'sufficiently' high activity with no specific pattern) could be permissive for the formation of callosal connections. In a recent study, we showed that callosal projection neurons preferentially make synaptic connections with neighboring callosal projection neurons and form local subnetworks (*Hagihara et al., 2021*). Some of the L events observed in this study may reflect the correlated subnetwork activity of callosal projection neurons, and such correlated activity may be critical for their axonal projections. This consideration may lead to a more general and intriguing idea that L events might be coactivations of locally connected cortical neurons, sending axons to the same target region(s). Future experiments would be able to test this important possibility by combining retrograde labeling of callosal (and other projection) neurons and Ca$^{2+}$ imaging in developing mice.

Network activity may also function to induce some trophic factors. If this is the case, the effect of reducing the activity of a small number of cells could be less than that of global activity reduction (*Mizuno et al., 2010*) because surrounding unperturbed cells can supply a 'trophic factor' for the perturbed cells. Spitzer et al. showed that BDNF could exert such a function in *Xenopus* spinal neurons (*Guemez-Gamboa et al., 2014*). They showed that spontaneous firing causes the release of BDNF, which non-cell-autonomously functions to trigger TrkB receptor activation and activity-dependent transmitter switching in the surrounding neurons. BDNF may function in callosal axon projections. BDNF expression is activity-dependent in the cortex (*Tao et al., 1998*; *Lein and Shatz, 2000*). It has also been shown that callosal axons require BDNF secretion for their projections (*Shimojo et al., 2015*). Activity-dependent secretion of BDNF, or other factor(s), may be associated with the network activity.

In the activity manipulation experiments using the Tet-off system (*Figures 1 and 4*), there may have been residual Kir2.1 expression after Dox treatment (*Figure 1—figure supplement 3*). Therefore, it remains uncertain precisely when activity levels returned and what patterns of spontaneous activity emerged during the Dox-treated period. Highly and/or less synchronous activity can be instructive or permissive to the formation of cortical circuits. As mentioned above, L events may be a mixture of different patterns of activity with mixed populations of neurons in the cortex. Some pattern(s) of network activity embedded in L events may play a role in the formation of callosal projections and cortical connections in general.

## Materials and methods

### Mice

The ICR strain of mice was used. In utero electroporation was performed in pregnant female mice, and their offspring (both male and female) were used in the study. All experiments were performed in accordance with the institutional animal welfare guidelines of the Animal Care and Use Committee of Kyoto University, Kyushu University, and Kagoshima University and were approved by the Committee of Animal Experimentation in the Graduate School of Science, Kyoto University (H2405, H2420, H2504, H2518), Graduate School of Medical and Dental Sciences, Kagoshima University (MD17080, MD21017), and the Ethical Committee of Kyushu University (A25-105-0).

### In utero electroporation and plasmids

In utero electroporation was performed as previously described (*Mizuno et al., 2007*: *Hagihara et al., 2015*). Briefly, pregnant mice were anesthetized at E15.5 with somnopentyl (pentobarbital sodium; 50 mg per kg of body weight; Kyoritsu-seiyaku) in saline with or without isoflurane. A midline laparotomy was performed to expose the uterus. For DNA microinjection, glass capillary tubes (GD-1; Narishige) were pulled using a micropipette puller (Sutter Instruments). Embryos were injected into the lateral ventricle with 1 µl of DNA solution (expression plasmids other than pCAG-tTA2$^s$, 0.6–1.0 mg ml$^{-1}$; pCAG-tTA2$^s$, 0.04–0.1 mg ml$^{-1}$), and square electric pulses (50 V; 50 ms) were delivered five times at the rate of one pulse per second using an electroporator (CUY21EDIT; NepaGene). After electroporation, the uterus was repositioned, the abdominal cavity was filled with prewarmed PBS, and the wall and skin were sutured. Animals were allowed to recover on a heating pad for approximately an hour before returning to their home cage. Plasmids used were as follows. pCAG-tTA2$^s$ and pTRETight2-Kir2.1 express Kir2.1 under the control of the Tet-off gene expression system (*Hagihara et al., 2015*). pCAsal-EGFP, pCAG-TurboRFP, and pCAG-FP635 express fluorescent proteins and pCAG-Kir2.1 for the expression of Kir2.1 under the control of the CAG promoter (*Mizuno et al., 2007*). pTRETight2-TurboRFP for expression of the fluorescent protein under the control of the Tet-off gene expression system. pTRETight2-Kir2.1EGFP (EGFP was fused with the C terminus of Kir2.1) was used to confirm the expression of Kir2.1. pCAG-hM3DGq for DREADD experiments. After birth, animals that did not show fluorescent signals in the visual cortex of the electroporated hemisphere were excluded from further histological or imaging experiments. For histological analyses, animals were anesthetized at P6, P8, P10, P15, P18, or P21 with an overdose of somnopentyl and decapitated.

### Doxycycline and clozapine-N-oxide administration

Dox (#9891: Sigma-Aldrich) was added to drinking water (2 mg ml$^{-1}$) with 10% sugar. Dox in water was protected from exposure to light and renewed every other day. Pups were exposed to Dox through the mother's milk; electroporated pups were transferred to a foster mother from P6 (or P9 or P12 in the experiments shown in *Figure 3*) that had been given Dox via the drinking water for a week. In the experiments shown in *Figure 2*, CNO (BML-NS105; Enzo Life Sciences; 1 mg ml$^{-1}$ in saline) was intraperitoneally injected every day. A previous study showed that an intraperitoneally injected CNO was effective (in terms of increasing activity) for about 9 hr (*Alexander et al., 2009*). The 'partial rescue' effect we observed (*Figure 2*) may suggest that activity was not fully restored during 24 hr by our daily CNO injections.

## Histology, confocal imaging, and quantification

Brains were fixed using 4% paraformaldehyde in PBS overnight and then transferred to 30% sucrose in PBS for 1–2 days. Coronal brain slices (50 μm) were sectioned using a freezing microtome (LM2000R; Leica). Fluorescent histological images were acquired using a confocal laser-scanning microscope (FV1000; Olympus). The expression of Kir2.1EGFP was assessed with anti-GFP antibody (rat mono-clonal Ab from Nacalai, Japan, GF090R; the secondary antibody, 488-anti Rat IgG from Invitrogen, A11006) (*Mizuno et al., 2007*). To obtain the projection patterns of fluorescently labeled callosal axons for each tissue section (50 μm in thickness), serial confocal images were collected at 3 μm intervals to create a z-axis image stack. Quantification of the callosal axon projections (*Figure 1D, E*) was performed as previously described with modifications (*Mizuno et al., 2007*). Briefly, in each brain section, boxes (300 μm in width, 100 μm in height) were drawn to enclose the fluorescently labeled callosal axon fibers in layers 1–3 and the WM on the projection side of the cortex, and the density of the fluorescent signal in each box was computed (these are raw values of the signal intensity in layers 1–3 and WM on the projection side). Then, a background level of signal was computed in adjacent areas of the cortex and subtracted (background-subtracted values of the signal intensity in layers 1–3 and WM on the projection side). To compensate for variability in the efficacy of labeling callosal axons with fluorescent proteins, the average WM fluorescent signal was obtained on the electroporated side of the cortex and used to normalize the signal intensity on the projection side of the cortex at each brain section (i.e. the normalized signal intensity in layers 1–3 and WM on the projection side was calculated by dividing the signal from layers 1–3 or WM on the projection side by the WM signal on the electroporated side; background-subtracted value of the signal intensity in layers 1–3 or WM on the projection side/background-subtracted value of the signal intensity in WM on the electroporated side). To quantify widths of callosal axon arborization in L2/3 and 5 (*Figure 1—figure supplement 4A, B*), confocal images were first Gaussian filtered (kernel size: 30 pixels) and then binarized by one of the authors who was blind to experimental conditions. Widths of positive regions that correspond to L2/3 and 5 arborizations in the binarized images were manually quantified. For densitometric line scans (*Figure 1—figure supplement 4C*), original confocal images were tilt-adjusted and cropped into 900 pixel × 900 pixel images so that pia to WM were covered. Pixel intensities of each row of the preprocessed images, which corresponds to a cortical depth, were averaged. For KirGFP expression level quantification (*Figure 1—figure supplement 3*), individual neurons were manually detected and rectangular ROIs were placed. Pixel intensities in green and red channels in each ROI were then averaged. For comparison across Dox treatment conditions, the ratio between the averaged green and red channel intensities from individual neurons was used (*Figure 1—figure supplement 3E*).

## Spontaneous neuronal activity recording and data analyses

Mice were prepared for in vivo calcium imaging as previously described (*Ohki et al., 2005*; *Hagihara et al., 2015*). In brief, mice were anesthetized using isoflurane (3.0% for induction, 1.0–2.0% during surgery). A custom-made metal plate was mounted onto the skull, and a craniotomy was carefully performed before calcium imaging approximately above V1 using stereotaxic coordinates. After surgery, the isoflurane concentration was reduced to 0.7%. Because the level of spontaneous activity is greatly affected by the anesthesia level (*Siegel et al., 2012*), the isoflurane concentration was carefully controlled, and we started recording at least 1 hr after the reduction in isoflurane concentration. We dissolved 0.8 mM Oregon Green 488 BAPTA-1 AM (OGB-1) in Dimethyl sulfoxide (DMSO) with 20% pluronic acid and mixed it with Artificial cerebrospinal fluid (ACSF) containing 0.05 mM Alexa594 (Alexa; all obtained from Invitrogen, CA, USA). A glass pipette (3–5 μm tip diameter) was filled with this solution and inserted into the cortex around the center of a craniotomy to a depth of approximately 250 μm from the surface, and then the solution was pressure-ejected from the pipette using a Picospitzer (Parker, USA) (−0.5 psi for 1–5 s, 5–10 times). After confirming loading, the craniotomy was sealed with a cover glass. We aimed to record spontaneous neuronal activity in putative binocular zones in V1 based on relative coordinates to lambda. Since the boundaries between V1 and higher visual areas, AL (anterolateral)/LM (lateromedial) are not as obvious as those in adult, our recordings likely contained lateral monocular V1 and AL/LM. Changes in calcium fluorescence in the cortical neurons were monitored using a two-photon microscope (Zeiss LSM7MP or Nikon A1MP), which was equipped with a mode-locked Ti:sapphire laser (MaiTai Deep See, Spectra Physics). The excitation light was focused with a 25× Olympus (NA: 1.05) or Nikon (NA: 1.10) PlanApo objective. The average

power delivered to the brain was <20 mW, depending on the depth of focus. OGB-1 and FP635 were excited at 920 nm. The emission filters were 517–567 nm for OGB-1 and 600–650 nm for FP635. Care was taken to shield the microscope objective and the photomultipliers from stray light. Images were obtained using Zeiss Zen software or Nikon NIS Elements software. A rectangular region (281.6 μm × 140.8 μm) from layer 2/3 (depths of 150–300 μm from the surface) was imaged with 512 × 256 pixels at 4 Hz.

Images were analyzed using custom-written in-house software running on in MATLAB (Mathworks) (*Ohki et al., 2005*; *Hagihara et al., 2015*; https://github.com/hagikent/CallosalRescue). Images were motion corrected by maximizing the correlation between frames. The cell outlines were automatically identified using template matching. The identified cell outlines were visually inspected, and the rare but clear errors were manually corrected. FP635 positive or negative neurons were manually identified. The time courses of individual cells were extracted by averaging the pixel values within the cell outlines, and then, high-cut (Butterworth, *n*=10; cutoff, 1 s) and low-cut (Gaussian, cutoff, 2 min) filters were applied. Those time courses were further corrected to minimize out-of-focus signal contamination. This process is important because of the highly synchronous characteristics of spontaneous network activity during the developmental stage. To achieve this, neuropil signals were subtracted from cell body signals after multiplying the contamination ratio as previously described (*Kerlin et al., 2010*; *Hagihara et al., 2015*). The corrected fluorescence signal from a cell body was estimated as follows:

$$\text{Fcell} - \text{corrected}(t) = \text{Fcell} - \text{apparent}(t) - r \times [\text{Fcell} - \text{surrounding}(t) - \text{mean}(\text{Fcell} - \text{surrounding})]$$

where t is the time and r is the contamination ratio. We calculated the contamination ratio r for each cell using the least squares method as follows:

$$\{F_{cell}[t_{base}] - \text{mean}[F_{cell}(t_{base})]\} = r \times \{F_{cell-surround}[t_{base}] - \text{mean}[F_{cell-surround}(t_{base})]\}$$

where $t_{base}$ is a period with no obvious spontaneous activity. After neuropil contamination correction, the spontaneous activity of each cell was detected using fixed criteria: ΔF/$F$=5%. Time periods when >20% cells were simultaneously active were regarded as synchronous spontaneous activity. This spontaneous activity was further classified into H events and L events (*Siegel et al., 2012*) based on the participation rate (H >60%; L: 60–20%). Note that we used slightly different criteria for H and L events because of our rigid neuropil subtraction methods.

## Statistical analyses

All data are expressed as individual data points and the mean ± SEM using a modified notBox-Plot function (originally written by R. Campbell), unless stated otherwise. Statistical analyses were conducted by Excel version 16.44 (Microsoft) (one-way ANOVA) and R version 2.12.0 (The R Foundation for Statistical Computing) (Tukey–Kramer test). Exact p-values and additional statistical information are provided in the 'Source data' (Supplementary file ). Throughout the study, p<0.05 was considered statistically significant. No statistical methods were used to pre-determine sample sizes, but our sample sizes (number of animal) are similar to those generally employed in the field.

## Acknowledgements

We thank T Kitazawa (FMI) for reading and commenting on the manuscript; T Murakami (U.Tokyo / Kyushu Univ.) for the help in calcium imaging data analysis validation; Y Hayashi (Tsukuba University) for the sequence information of Kir2.1EGFP; All the members of Tagawa, Hirano, and Ohki laboratories for discussion. Core Research for Evolutionary Science and Technology (CREST) - Japan Science and Technology Agency (JST) (to KO and Y Tagawa); Brain Mapping by Integrated Neurotechnologies for Disease Studies (Brain/MINDS)-Japan Agency for Medical Research and Development (AMED) (to KO); Institute for AI and Beyond (to KO); Japan Society for Promotions of Sciences (JSPS) KAKENHI (Grant numbers 25221001, 25117004, 19H01006, and 19H05642 to KO; 23500388 and 16 K06992, and 21K06374 to Y Tagawa); "Neural Diversity and Neocortical Organization" (23123508 and 25123707 to Y Tagawa); "Dynamic Regulation of Brain Function by Scrap & Build System" (17H05745 and 19H04756 to Y Tagawa); Astellas Foundation for Research on Metabolic Disorders (to Y Tagawa); The Kodama Memorial Fund for Medical Research (to Y Tagawa); The Novartis Foundation (Japan)

for the Promotion of Science (to Y Tagawa); The Uehara Memorial Foundation (to Y Tagawa); Takeda Science Foundation (to KMH).

## Additional information

### Funding

| Funder | Grant reference number | Author |
|---|---|---|
| Core Research for Evolutional Science and Technology | | Yoshiaki Tagawa Kenichi Ohki |
| Japan Agency for Medical Research and Development | | Kenichi Ohki |
| Institute for AI and Beyond | | Kenichi Ohki |
| Japan Society for the Promotion of Science | 25221001 | Kenichi Ohki |
| Japan Society for the Promotion of Science | 25117004 | Kenichi Ohki |
| Japan Society for the Promotion of Science | 19H01006 | Kenichi Ohki |
| Japan Society for the Promotion of Science | 19H05642 | Kenichi Ohki |
| Japan Society for the Promotion of Science | 16 K06992 | Yoshiaki Tagawa |
| Japan Society for the Promotion of Science | 21K06374 | Yoshiaki Tagawa |
| Neural Diversity and Neocortical Organization | 23123508 and 25123707 | Yoshiaki Tagawa |
| Dynamic Regulation of Brain Function by Scrap & Build System | 17H05745 | Yoshiaki Tagawa |
| Dynamic Regulation of Brain Function by Scrap & Build System | 19H04756 | Yoshiaki Tagawa |
| Astellas Foundation for Research on Metabolic Disorders | | Yoshiaki Tagawa |
| The Kodama Memorial Fund for Medical Research | | Yoshiaki Tagawa |
| NOVARTIS Foundation (Japan) for the Promotion of Science | | Yoshiaki Tagawa |
| Uehara Memorial Foundation | | Yoshiaki Tagawa |
| Takeda Science Foundation | | Kenta M Hagihara |

The funders had no role in study design, data collection and interpretation, or the decision to submit the work for publication.

### Author contributions

Yuta Tezuka, Conceptualization, Data curation, Investigation; Kenta M Hagihara, Conceptualization, Data curation, Software, Formal analysis, Investigation, Visualization, Methodology, Writing – original draft, Writing – review and editing; Kenichi Ohki, Supervision, Funding acquisition, Writing – review

and editing; Tomoo Hirano, Supervision, Project administration, Writing – review and editing; Yoshiaki Tagawa, Conceptualization, Data curation, Supervision, Funding acquisition, Investigation, Writing – original draft, Project administration, Writing – review and editing

### Author ORCIDs
Kenta M Hagihara ![ORCID] http://orcid.org/0000-0003-0064-0852
Tomoo Hirano ![ORCID] http://orcid.org/0000-0003-3685-5935
Yoshiaki Tagawa ![ORCID] http://orcid.org/0000-0002-3300-090X

### Ethics
All experiments were performed in accordance with the institutional animal welfare guidelines of the Animal Care and Use Committee of Kyoto University, Kyushu University, and Kagoshima University and were approved by the Committee of Animal Experimentation in the Graduate School of Science, Kyoto University (H2405, H2420, H2504, H2518), Graduate School of Medical and Dental Sciences, Kagoshima University (MD17080, MD21017), and the Ethical Committee of Kyushu University (A25-105-0).

### Decision letter and Author response
Decision letter https://doi.org/10.7554/eLife.72435.sa1
Author response https://doi.org/10.7554/eLife.72435.sa2

## Additional files

### Supplementary files
• Transparent reporting form

### Data availability
All data generated or analysed during this study are included in the manuscript and supporting file; Source Data files have been provided for Figures 1-4. All datasets associated with this manuscript are uploaded in Dryad: https://doi.org/10.5061/dryad.18931zcz0.

The following dataset was generated:

| Author(s) | Year | Dataset title | Dataset URL | Database and Identifier |
|---|---|---|---|---|
| Tagawa Y | 2022 | eLife20210804a | https://dx.doi.org/10.5061/dryad.18931zcz0 | Dryad Digital Repository, 10.5061/dryad.18931zcz0 |

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
