## [Editor Report]

This study extends previous findings on the activity-dependent development of the long-range axonal connections between the cortical hemispheres. In particular, it shows that neuronal activity strongly influences the development of these axons during a particular developmental period and it provides convincing evidence that specific activity patterns are required for the proper development of these long-range connections.

---

## [Decision Letter]

**Decision letter after peer review:**

Thank you for submitting your article "Developmental stage-specific patterned activity contributes to callosal axon projections" for consideration by *eLife*. Your article has been reviewed by 3 peer reviewers, and the evaluation has been overseen by a Reviewing Editor and Catherine Dulac as the Senior Editor. The following individuals involved in review of your submission have agreed to reveal their identity: Edward S Ruthazer (Reviewer #1); Christian Albrecht Lohmann (Reviewer #2).

Essential revisions:

(1) Some independent evidence that the presence of DOX reduces KIR expression.

(2) Improved characterization of the callosal projections.

(3) A revision in which the authors make clear that the current data does not support a role for patterned activity. Reviewers are also open to new experiments (suggested by Reviewer 2) but it is acknowledged that these are challenging experiments, but if possible, would greatly enhance the impact of the paper.

*Reviewer #1 (Recommendations for the authors):*

What would strengthen the paper considerably would be to use calcium imaging to better characterize the restoration of activity during the critical period rather than just at the end both as a validation of the tet-off model and to precise the actual critical period window. In addition, more quantitative description of the actual size and shape, and not just the intensity, of the callosal projection over numerous rostrocaudal points would provide a better understanding of the importance of activity.

Abstract

Line 33 of abstract "less correlated network activity was critical"

Intro

Line 41 Katz and Shatz review is quite out of date already. Can it be replaced with something more recent?

Line 52 the Smith et al. 2018 paper from David Fitzpatrick could also be referenced here in the context of corticocortical spontaneous activity.

Line 59 Following a lengthy list of studies on how patterned activity influence cortical circuit development you go on to state "However their role in cortical circuit formation remains elusive." Is it your intention to be so dismissive of the studies you just cited? Perhaps it would be more precise to say "However, their role in many aspects of cortical circuit formation remains to be fully elucidated."

Line 61 "THE callosal projection…".

Overall, the intro is well written and a nice summary of the field to date with just a few missing references.

Results

Line 107 "the projection pattern was quite similar".

Line 119 "quantitative analyses suggested that the extent of callosal axons… was comparable".

I didn't see any analysis of the EXTENT of callosal axons. Can this be quantified by measuring ML extent, etc?

Line 114 starting dox on P6 with the hope that Kir 2.1 will be gone from P10-14 seems to require validation either by western blot or electrophysiology or functional imaging.

The body of text associated with figure 4 mentions the use of FP635 but fails to mention how calcium was measured (OGB1). This is a fairly major oversight.

Line 180 "This implies that L events, which have a less synchronous pattern of network activity, make a larger contribution to the formation of callosal axon projections". I don't believe this is an accurate description. L events contribute to the formation of callosal projections but their relative importance is not possible to ascertain from this data. Rather one can conclude that any form of activity, synchronous or not, appears to be permissive for the presence of late stage callosal connections. While some useful information may be gleaned from these data it is difficult to directly attribute activity patterns at P13 to the formation of callosal connectivity that has taken place over the preceding week. What this data is particularly useful for is to establish that Dox does partially restore activity. Therefore, it would have been even more useful to have examined the relative proportions of L vs H activity at or BEFORE P6 in control animals to understand why P6 represents the opening of the critical period and to examine at several times between P6 and P13 the restoration of activity by Dox treatment to understand when the Kir2.1 expression in fact is sufficiently reduced to permit normal activity.

Further, the question of interhemispheric synchrony seems potentially relevant though it is not touched upon at all in these experiments.

Discussion

Line 191 and 197 "Our findings add new evidence … long-range axonal projections require patterned spontaneous activity for their normal development." Nothing in this study distinguishes patterned activity from random activation. These statements should be removed.

Line 228 includes a reference to "data not shown". Please remove this sentence or provide a supplementary figure. It is a fascinating point which would be good to elaborate upon but perhaps the experiments are insufficient to draw firm conclusions yet, in which case they should not even be mentioned.

Line 232 states "the rescue effect was only observed during P10-15". What is the basis for extrapolating from P6 to P10? This is a very precise statement which is not well supported by the data. We only know about P6 to P15.

Methods

Line 333 custom software for Ca analysis should be made available in a public repository or uploaded with the manuscript. The dryad reference should probably be added to the methods?

Lines 807-10: the normalization of signal intensity is somewhat vague, but absolutely essential. When is wm signal used and when is layer 1-3 signal used for normalization.

Figures

Bar graphs showing mean {plus minus} SEM fail to convey the real variability of the data. If possible consider adding the individual points that make up the data to the graphs, even though these are available in the supplemental data.

Suppl Figure 1B shows a different coronal plane than the other images. Does this indicate a different rostrocaudal pattern of interhemispheric connectivity in the rescue animals from controls? If not can a properly matched section be provided, or better yet a series throughout the rostrocaudal length of V1?

Suppl Figure 2 is not cited anywhere in the manuscript. Note that expression of a soluble FP cannot be used as a proxy for expression of a transmembrane Kir2.1

*Reviewer #2 (Recommendations for the authors):*

The authors propose the "intriguing idea that L events might be coactivations of locally connected cortical neurons, sending axons to the same target region(s)" (Lines 216-218). It would be interesting to see whether some L events indeed co-activate neurons projecting to the same target regions, by retrogradely labeling e.g. callosally projecting cortical neurons and performing calcium imaging to identify activity events where these subpopulations of neurons participate specifically.

---

## [Author Response]

Essential revisions:(1) Some independent evidence that the presence of DOX reduces KIR expression.(2) Improved characterization of the callosal projections.(3) A revision in which the authors make clear that the current data does not support a role for patterned activity. Reviewers are also open to new experiments (suggested by Reviewer 2) but it is acknowledged that these are challenging experiments, but if possible, would greatly enhance the impact of the paper.

We thank the editor and the reviewers for their constructive comments and valuable suggestions. Based on the comments, we have revised the manuscript extensively. Regarding the “Essential Revisions” #1 and #2, we have conducted additional experiments/analyses and added new results (Figure 1—figure supplement 3 and 4). Regarding the “Essential Revisions” #3, we added Discussions in the manuscript.

Regarding the “Essential Revisions” #1, we conducted new experiments using pTRE-Tight2Kir2.1EGFP, with which EGFP signals reflect localization of over-expressed Kir2.1, and examined when the expression of Kir2.1EGFP went down after Dox treatment at P6. At P6 (before Dox treatment), the signals of Kir2.1EGFP (stained with anti-GFP antibody) were observed in the periphery of the soma and along dendrites, implying that Kir2.1EGFP was transported to the cellular membrane. At P10 and P15 (4 days and 9 days after Dox treatment), Kir2.1EGFP signals were not observed in the periphery of the soma and along dendrites. We noted that low-level green signals were observed in the central part of the cell body. These may stem from low-level expression of Kir2.1EGFP in nuclei or cytosol even after Dox treatment. Alternatively, and more likely, these may reflect bleed-through of RFP signals into GFP channel. Overall, we confirmed that Kir2.1 proteins that were localized to the cellular membrane were largely down-regulated. We described these observations in detail in the figure legend of Figure 1—figure supplement 3, and added the result as Figure 1—figure supplement 3.

Regarding the “Essential Revisions” #2, we added analyses of the pattern of callosal projections; the width of callosal axon innervation zone in layers 2/3 and 5, and densitometric line scans across all cortical layers. We added these results as Figure 1—figure supplement 4.

Regarding the “Essential Revisions” #3, we removed the term “patterned activity” throughout the manuscript and revised the title, abstract, introduction, results, and discussion extensively. The suggested new experiment (suggested by Reviewer #2), retrograde labeling of callosal projection neurons and performing in vivo Ca^2+^ imaging to investigate to what extent retrogradely labeled callosal projection neurons participate in each L event, is a key to test the hypothesis. This experiment can be conducted easily in the mature brain, but difficult in the neonatal period. Firstly, retrograde labeling takes several days from dye injection to fully labeling of target neurons; If we want to perform Ca^2+^ imaging at P12, we need to inject dye around P9 when callosal axons do not fully reach and ramify in the contralateral cortex. Secondly, it is desired to label almost 100% of callosal projection neurons in this experiment. L events are network activity with intermediate-level synchrony (some neurons are participated but many other neurons are not), and it would be difficult to determine quantitatively to what extent the two populations of neurons, retrogradely labeled callosal projection neurons and neurons that participate in each L event, are overlapping if retrograde labeling is not perfect. Because the neonatal period is when the formation of callosal projections is still ongoing, complete retrograde labeling is difficult. Another way to label callosal projection neurons is genetic labeling, but we havenʼt found a perfect genetic tool to specifically label callosal projection neurons. We have conducted preliminary experiments, but it will take more time and is too hard to complete within a reasonable time span. We therefore revised the text in the Discussion as follows.

Page 8, line 228-231:

“This consideration may lead to a more general and intriguing idea that L events might be coactivations of locally connected cortical neurons, sending axons to the same target region(s). Future experiments would be able to test this important possibility by combining retrograde labeling of callosal (and other projection) neurons and Ca^2+^ imaging in developing mice.”

Page 8, line 241-244:

“Highly and/or less synchronous activity can be instructive or permissive to the formation of cortical circuits. As mentioned above, L events may be a mixture of different patterns of activity with mixed populations of neurons in the cortex. Some pattern(s) of network activity embedded in L events may play a role in the formation of callosal projections and cortical connections in general.”

We respond to each reviewerʼs comment one-by-one as follows.

Reviewer #1 (Recommendations for the authors):What would strengthen the paper considerably would be to use calcium imaging to better characterize the restoration of activity during the critical period rather than just at the end both as a validation of the tet-off model and to precise the actual critical period window. In addition, more quantitative description of the actual size and shape, and not just the intensity, of the callosal projection over numerous rostrocaudal points would provide a better understanding of the importance of activity.

We thank the reviewer for these constructive comments. Repeated calcium imaging during the critical period and analysis of the pattern of callosal projections over numerous rostrocaudal points are too hard to complete within a reasonable time span, but these would be valuable experiments to conduct in future study.

AbstractLine 33 of abstract "less correlated network activity was critical"

We revised the abstract as follows.

“Furthermore, in vivo Ca^2+^ imaging revealed that the projections could be established even without fully restoring highly synchronous activity.”

IntroLine 41 Katz and Shatz review is quite out of date already. Can it be replaced with something more recent?

We respectfully disagree that the review by Katz and Shatz is outdated, but we view it is still foundational. Nevertheless, we agree that adding a newer review that encompasses recent progress as well would be more informative, and thus, additionally cited Ackman and Crair (Page 3, line 42)

Line 52 the Smith et al. 2018 paper from David Fitzpatrick could also be referenced here in the context of corticocortical spontaneous activity.

Thank you for this suggestion. We have added this reference (Page 3, line 52).

Line 59 Following a lengthy list of studies on how patterned activity influence cortical circuit development you go on to state "However their role in cortical circuit formation remains elusive." Is it your intention to be so dismissive of the studies you just cited? Perhaps it would be more precise to say "However, their role in many aspects of cortical circuit formation remains to be fully elucidated."Line 61 "THE callosal projection…".

Thank you for the valuable suggestions. We have revised the text as suggested.

Overall, the intro is well written and a nice summary of the field to date with just a few missing references.ResultsLine 107 "the projection pattern was quite similar".Line 119 "quantitative analyses suggested that the extent of callosal axons… was comparable".I didn't see any analysis of the EXTENT of callosal axons. Can this be quantified by measuring ML extent, etc?

Based on these comments, we added analyses of the pattern of callosal projections; the width of callosal axon innervation zone in layers 2/3 and 5, and densitometric line scans across all cortical layers. We added these results as Figure 1—figure supplement 4.

Line 114 starting dox on P6 with the hope that Kir 2.1 will be gone from P10-14 seems to require validation either by western blot or electrophysiology or functional imaging.

As mentioned, we conducted new experiments using pTRE-Tight2-Kir2.1EGFP, and examined when the expression of Kir2.1EGFP went down after Dox treatment at P6 by immunohistochemistry. We added the result as Figure 1—figure supplement 3 and described observations in detail in the figure legend of Figure 1—figure supplement 3.

The body of text associated with figure 4 mentions the use of FP635 but fails to mention how calcium was measured (OGB1). This is a fairly major oversight.

Thank you for pointing this out. We have added this critical information in the text.

Line 180 "This implies that L events, which have a less synchronous pattern of network activity, make a larger contribution to the formation of callosal axon projections". I don't believe this is an accurate description. L events contribute to the formation of callosal projections but their relative importance is not possible to ascertain from this data. Rather one can conclude that any form of activity, synchronous or not, appears to be permissive for the presence of late stage callosal connections. While some useful information may be gleaned from these data it is difficult to directly attribute activity patterns at P13 to the formation of callosal connectivity that has taken place over the preceding week. What this data is particularly useful for is to establish that Dox does partially restore activity. Therefore, it would have been even more useful to have examined the relative proportions of L vs H activity at or BEFORE P6 in control animals to understand why P6 represents the opening of the critical period and to examine at several times between P6 and P13 the restoration of activity by Dox treatment to understand when the Kir2.1 expression in fact is sufficiently reduced to permit normal activity.Further, the question of interhemispheric synchrony seems potentially relevant though it is not touched upon at all in these experiments.

We thank the reviewer for these constructive comments. Repeated calcium imaging during the critical period, as well as an investigation of the role of interhemispheric synchrony, is too hard to complete within a reasonable time span, but these are surely valuable experiments to conduct in the next step after this study.

DiscussionLine 191 and 197 "Our findings add new evidence … long-range axonal projections require patterned spontaneous activity for their normal development." Nothing in this study distinguishes patterned activity from random activation. These statements should be removed.

Based on this comment, we removed the term “patterned activity” throughout the manuscript and revised the text.

Line 228 includes a reference to "data not shown". Please remove this sentence or provide a supplementary figure. It is a fascinating point which would be good to elaborate upon but perhaps the experiments are insufficient to draw firm conclusions yet, in which case they should not even be mentioned.

Based on this comment, we removed the reference “data not shown” and revised the text as follows.

Page 8, lines 232-240

“Network activity may also function to induce some trophic factors. If this is the case, the effect of reducing the activity of a small number of cells could be less than that of global activity reduction (Mizuno et al., 2010) because surrounding unperturbed cells can supply a “trophic factor” for the perturbed cells. Spitzer et al. showed that BDNF could exert such a function in *Xenopus* spinal neurons (Guemez-Gamboa et al., 2014). They showed that spontaneous firing causes the release of BDNF, which non-cell-autonomously functions to trigger TrkB receptor activation and activity-dependent transmitter switching in the surrounding neurons. BDNF may function in callosal axon projections. BDNF expression is activity-dependent in the cortex (Tao et al., 1998; Lein and Shatz, 2000). It has also been shown that callosal axons require BDNF secretion for their projections (Shimojo et al., 2015). Activity-dependent secretion of BDNF, or other factor(s), may be associated with the network activity.”

Line 232 states "the rescue effect was only observed during P10-15". What is the basis for extrapolating from P6 to P10? This is a very precise statement which is not well supported by the data. We only know about P6 to P15.

In response to this comment, we revised the text accordingly.

MethodsLine 333 custom software for Ca analysis should be made available in a public repository or uploaded with the manuscript. The dryad reference should probably be added to the methods?

In response to this comment, we made analysis codes publicly available at Github and updated the method section as follows.

“Images were analyzed using custom codes written in MATLAB (Mathworks) (Ohki et al., 2005; Hagihara et al., 2015; https://github.com/hagikent/CallosalRescue).”

Lines 807-10: the normalization of signal intensity is somewhat vague, but absolutely essential. When is wm signal used and when is layer 1-3 signal used for normalization.

In response to this comment, we revised the method section as follows.

“Quantification of the callosal axon projections (Figure 1D and E) was performed as previously described with modifications (Mizuno et al., 2007). Briefly, in each brain section, boxes (300 micrometers in width, 100 micrometers in height) were drawn to enclose the fluorescently labeled callosal axon fibers in layers 1-3 and the white matter (WM) on the projection side of the cortex, and the density of the fluorescent signal in each box was computed (these are raw values of the signal intensity in layers 1-3 and WM on the projection side). Then, a background level of signal was computed in adjacent areas of the cortex and subtracted (background-subtracted values of the signal intensity in layers 1-3 and WM on the projection side). To compensate for variability in the efficacy of labeling callosal axons with fluorescent proteins, the average WM fluorescent signal was obtained on the electroporated side of the cortex and used to normalize the signal intensity on the projection side of the cortex at each brain section (i.e., the normalized signal intensity in layers 1-3 and WM on the projection side was calculated by dividing the signal from layers 1-3 or WM on the projection side by the WM signal on the electroporated side; background-subtracted value of the signal intensity in layers 1-3 or WM on the projection side / background-subtracted value of the signal intensity in WM on the electroporated side).”

FiguresBar graphs showing mean {plus minus} SEM fail to convey the real variability of the data. If possible consider adding the individual points that make up the data to the graphs, even though these are available in the supplemental data.

In response to this critical comment, we revised the Figures as suggested.

Suppl Figure 1B shows a different coronal plane than the other images. Does this indicate a different rostrocaudal pattern of interhemispheric connectivity in the rescue animals from controls? If not can a properly matched section be provided, or better yet a series throughout the rostrocaudal length of V1?

We did not mean a different rostrocaudal pattern of interhemispheric connectivity in the rescue animals. The pattern of callosal projections does not change very much from section to section in the rostrocaudal axis in the primary visual cortex (for the reference, please see Figure 1D of Mizuno et al., JNS, 27, 6760-6770, 2007). Although the 4 sections shown in Figure1—figure supplement 2 are slightly variable in the rostrocaudal axis, we believe these figures illustrate that callosal projections are clearly formed in E15-P15 DOX condition (B) just as normal animals and that Kir2.1 expression (by pCAG vector or Tet-OFF system/no DOX condition) impairs callosal projections (C, D).

Suppl Figure 2 is not cited anywhere in the manuscript. Note that expression of a soluble FP cannot be used as a proxy for expression of a transmembrane Kir2.1

As mentioned, we conducted new experiments using pTRE-Tight2-Kir2.1EGFP, and examined when the expression of Kir2.1EGFP went down after Dox treatment at P6 by immunohistochemistry. We added the result as Figure 1—figure supplement 3 and described observations in detail in the figure legend of Figure 1—figure supplement 3.

Reviewer #2 (Recommendations for the authors):The authors propose the "intriguing idea that L events might be coactivations of locally connected cortical neurons, sending axons to the same target region(s)" (Lines 216-218). It would be interesting to see whether some L events indeed co-activate neurons projecting to the same target regions, by retrogradely labeling e.g. callosally projecting cortical neurons and performing calcium imaging to identify activity events where these subpopulations of neurons participate specifically.

We thank the reviewer for this fascinating suggestion. The suggested experiment, retrograde labeling of callosal projection neurons and performing in vivo Ca^2+^ imaging to investigate to what extent retrogradely labeled callosal projection neurons participate in each L event, is a key to test the hypothesis. This experiment can be conducted easily in the mature brain, but difficult in the neonatal period. Firstly, retrograde labeling takes several days from dye injection to fully labeling of target neurons; If we want to perform Ca^2+^ imaging at P12, we need to inject dye around P9 when callosal axons do not fully reach and ramify in the contralateral cortex. Secondly, it is desired to label almost 100% of callosal projection neurons in this experiment. L events are network activity with intermediate-level synchrony (some neurons are participated but many other neurons are not), and it would be difficult to determine quantitatively to what extent the two populations of neurons, retrogradely labeled callosal projection neurons and neurons that participate in each L event, are overlapping if retrograde labeling is not perfect. Because the neonatal period is when the formation of callosal projections is still ongoing, complete retrograde labeling is difficult. Another way to label callosal projection neurons is genetic labeling, but we havenʼt found a perfect genetic tool to specifically label callosal projection neurons. We have conducted preliminary experiments, but it will take more time and is too hard to complete within a reasonable time span. We therefore revised the text in the Discussion as follows.

Page 8, line 228-231:

“This consideration may lead to a more general and intriguing idea that L events might be coactivations of locally connected cortical neurons, sending axons to the same target region(s). Future experiments would be able to test this important possibility by combining retrograde labeling of callosal (and other projection) neurons and Ca^2+^ imaging in developing mice.”

Page 8, line 241-244:

“Highly and/or less synchronous activity can be instructive or permissive to the formation of cortical circuits. As mentioned above, L events may be a mixture of different patterns of activity with mixed populations of neurons in the cortex. Some pattern(s) of network activity embedded in L events may play a role in the formation of callosal projections and cortical connections in general.”